# BmTudor-sn Is a Binding Protein of Destruxin A in Silkworm Bm12 Cells

**DOI:** 10.3390/toxins11020067

**Published:** 2019-01-24

**Authors:** Jingjing Wang, Weina Hu, Qiongbo Hu

**Affiliations:** Key Laboratory of Bio-Pesticide Innovation and Application of Guangdong Province, College of Agriculture, South China Agricultural University, Guangzhou 510642, China; wangjingjing@stu.scau.edu.cn (J.W.); hwn688094@163.com (W.H.)

**Keywords:** Destruxin A, *Bombyx mori*, binding protein, BmTudor-sn, Bm12 cell

## Abstract

Destruxin A (DA), a hexa-cyclodepsipeptidic mycotoxin secreted by the entomopathogenic fungus *Metarhizium anisopliae*, was reported to have an insecticidal effect and anti-immunity activity. However, its molecular mechanism of action remains unclear. Previously, we isolated several potential DA-affinity (binding) proteins in the *Bombyx mori* Bm12 cell line. By docking score using MOE2015, we selected three proteins—BmTudor-sn, BmPiwi, and BmAGO2—for further validation. First, using Bio-Layer Interferometry in vitro, we found that BmTudor-sn had an affinity interaction with DA at 125, 250, and 500 µM, while BmPiwi and BmAGO2 had no interaction signal with DA. Second, we employed standard immunoblotting to verify that BmTudor-sn is susceptible to DA, but BmPiwi and BmAGO2 are not. Third, to verify these findings in vivo, we used a target engagement strategy based on shifts in protein thermal stability following ligand binding termed the cellular thermal shift assay and found no thermal stability shift in BmPiwi and BmAGO2, whereas a shift was found for BmTudor-sn. In addition, in BmTudor-sn knockdown Bm12 cells, we observed that cell viability increased under DA treatment. Furthermore, insect two-hybrid system results indicated that the key site involved in DA binding to BmTudor-sn was Leu704. In conclusion, in vivo and in vitro experimental evidence indicated that BmTudor-sn is a binding protein of DA in silkworm Bm12 cells at the 100 µM level, and the key site of this interaction is Leu704. Our results provide new perspectives to aid in elucidating the molecular mechanism of action of DA in insects and developing new biopesticide.

## 1. Introduction

Destruxins are cyclodepsipeptidic mycotoxins, and there are 39 analogues [1]. Destruxin A (DA, Figure 1A), the common analogue secreted by the entomopathogenic fungus *Metarhizium anisopliae*, has a strong insecticidal effect and anti-immunity activity, which includes breaking the balance between calcium and hydrogen ions and subsequently affecting the function of phagocytosis and encapsulation in hemocyte [2,3]. Because of the immunosuppression activity of DA in insects, the majority of studies have focused on the mechanism of action of the effect of DA on the immune-related pathway or stress reaction (response), such as changes to the transcriptome [4] and proteome [5] or immune regulation by microRNA [6] in *Plutella xylostella*. There have been various studies on the effects of DA, including its impact on transcriptome [7], proteome, transcription factor, and antibacterial peptide expression [8] in *Bombyx mori* and its influence on the Toll or Imd pathway in *Bemisia tabaci* [9]. However, for drug research or development, it is important to look for direct targets, most of which are proteins, in appropriate tissues or cells. In order to clarify the molecular mechanism more directly, previously, we screened and isolated DA-binding proteins in ovary-derived Bm12 cells using a label-free small molecule drug target identification method called drug affinity responsive target stability (DARTS) [10]. The DARTS results indicated that DA greatly induced heat shock proteins (HSPs) in cultured cells, and we successfully demonstrated that DA binds to an HSP in vitro by molecular interaction validation [11]. However, evidence from only one approach is not sufficient, especially for studies only performed in vitro.

Here, we selected three proteins, BmTudor-sn, BmPiwi, and BmAGO2, from the DARTS analysis based on their molecular docking score and determined whether these proteins are DA binding proteins through a series of in vivo and in vitro experiments. In brief, BmPiwi and BmAGO2 belong to the Argonaute family [12], which plays a vital role in germ cell development and represent a core component of the RNA interference (RNAi) pathway and function in transcriptional regulation [13]. Meanwhile, BmTudor-sn, a multifunctional protein containing four staphylococcal nuclease domains and a Tudor domain, participates in cellular pathways involved in gene regulation, cell growth, and development and interacts with Argonaute proteins [14], and it is known as stress granule protein in *Bombyx mori* [15]. These three proteins are all critical components in the RNAi pathway [16]. Notably, few studies have associated DA with germ line-derived proteins that are involved in several important physiological processes. In addition, this study may provide novel insights to exploit new targets and pathways for the development of pesticides.

## 2. Results

### 2.1. Molecular Docking

Faced with hundreds of isolated proteins from the DARTS analysis, molecular docking was a convenient and efficient method to screen for proteins that are of research value. Ultimately, BmTudor-sn, BmPiwi, and BmAGO2 protein were selected from hundreds of candidates because of their relatively high binding scores and because they play crucial roles in many physiological processes in *Bombyx mori*.

Through molecular docking, we obtained the binding modes of DA with the BmTudor-sn, BmPiwi, and BmAGO2 proteins. The estimated binding free energies indicated by GBVI/WSA dG scoring are listed in Table 1. A lower binding free energy suggests a higher binding affinity. The ligands and DA had docking scores that ranged from −9.1947 to −11.002 kcal/mol, suggesting a good binding affinity with these proteins. The binding modes of DA with BmTudor-sn, BmPiwi, and BmAGO2 are depicted in Figure 1. DA fits the pocket well in terms of the shape in the binding sites. For the BmTudor-sn docking pattern, the Arg343 side chain of the SN3 domain could have a hydrogen bond interaction with the carbonyl group of DA, which contains a number of carbonyl groups that might form hydrogen bonds with nearby basic amino acids. For the BmPiwi binding mode, the carbonyl in DA, which is regarded as a hydrogen bond donor, forms one hydrogen bond with the side chain of Cys648 in BmPiwi, which forms other hydrogen bonds with Gln695 and Gly651. At the bottom of the binding pocket, hydrogen bonds are formed between Cys614 and several residues around it. For the binding mode of DA with BmAGO2, one oxygen atom of the carbonyl near the pyrrolidine of DA, which is regarded as a hydrogen bond donor, forms one hydrogen bond with the side chain of Arg981 in BmAGO2 and forms another intramolecular hydrogen bond with a nitrogen atom. The other oxygen atom of the carbonyl near the pyrrolidine of DA, which is regarded as a hydrogen bond donor, forms one hydrogen bond with the side chain of Lys761 in BmAGO2.

### 2.2. Assessing the Interaction of DA with BmTudor-sn, BmPiwi, and BmAGO2 by Bio-Layer Interferometry (BLI) In Vitro

The determination of the affinity constant between heterologously expressed recombinant proteins with a small molecule through in vitro biophysical approaches is a generally accepted and frequently used method for target validation. In this study, we selected bio-layer interferometry [17] because it is label free, has high sensitivity, and can be performed in real-time to assess the binding affinity. Recombinant proteins were expressed and purified from eukaryotic expression in lepidoptera *Spodoptera frugiperda* 9 (Sf9) cells because of their similar tertiary structure with native proteins. The BLI analysis results indicated that DA interacts with BmTudor-sn at concentrations of 125 µm, 250 µm, and 500 µm but not BmPiwi or BmAGO2 (Figure 2A,B). And the affinity constant KD is 5.87 × 10^−4^M. These results possibly contradict the above docking results, which indicated that DA formed hydrogen bonds with the three proteins.

### 2.3. Assessing the Interaction of DA with BmTudor-sn through Protein Stability and RNAi In Vivo

DARTS is based on the principle that protein folding stability is more resistant to protease and heat treatments under conformational modifications caused by small molecules or other ligands. Notably, false-positive DARTS results will be obtained with the high expression of proteins caused by drug treatment and the stress response thus, it was necessary to confirm that BmTudor-sn was not completely hydrolyzed by proteolysis. The immunoblotting results (Figure 3A) clearly indicated that BmTudor-sn, but not BmPiwi and BmAGO2, was stable with DA treatment in a dosage-dependent manner under proteolysis, which suggested that BmTudor-sn is a DA binding protein. Additional evidence was obtained using a method that directly monitored target engagement based on the shift in protein thermal stability induced by a small molecule, which is termed the cellular thermal shift assay (CETSA) [18]. As depicted in Figure 3B, the thermal stability of BmTudor-sn increased following DA binding modification, as indicated by the binding protein solubility in the supernatant being positively correlated with the temperature gradient. These protein stability shift assay results provided sufficient evidence to demonstrate that BmTudor-sn is a binding protein of DA in Bm12 cells. Then, cytotoxicity and viability assays were performed following DA treatment with BmTudor-sn knockdown, and the results revealed that viability increased by approximately 20% (Figure 3C).

### 2.4. Screening Key Amino Acid Sites of Interaction between DA and BmTudor-sn Using an Insect Two-Hybrid (I2H) System

The above experiments assessed and verified the interaction of DA with BmTudor-sn. However, the exact interaction sites were not clear. Here, we evaluated the binding site in vivo using the insect two-hybrid (I2H) protein-protein interaction method [19] (Figure 4). This method was used to assess key sites where DA disrupted the interaction of BmTudor-sn with BmAGO1 in the Spodoptera frugiperda 9 (Sf9) cell line. Six mutants were prepared according to optimal docking sites, and 0.02 and 0.2 µg/mL DA treatments were selected based on our previous results. The results clearly indicated that for the 0.02 and 0.2 µg/mL treatments, the signals for 2 mutants, Ser707Ala and leu704Ala, and 3 mutants, Ser701Ala, Leu704Ala, and Tyr708Ala, respectively, were each different from the wild type signal, which suggested that the binding mode is dependent on the DA dosage. Apparently, DA interacts with BmTudor-sn at both concentrations at Leu704. At the protein domain level, the differences in the mutants were both in the Tudor domain (Ser707, Leu704, Ser701, Tyr708) rather than the SN3 (Lys492) and SN4 (Lys582) domains. In addition, DA impeded the protein pair interaction at Ser707 and Tyr708 and promoted it at Leu704 and Ser701 based on the increase or decrease in the signal, respectively.

## 3. Discussion

In drug research and development, it is important to clearly determine the mechanism of action of small molecules in cells, the most common target of which are binding proteins [20]. Previously, we attempted to elucidate the mechanism for DA using different approaches, such as analyzing phenotype differences and changes in the transcriptome^7^ and proteome [21]. Unfortunately, because DA has no active group, it is not possible to bond affinity chromatography tags for use in chemical proteomics. With DARTS, we have the ability to bond DA with potential proteins. However, DARTS can return false-positive results when highly expressed proteins are not digested completely by protease [22]. Indeed, BmTudor-sn, BmPiwi, and BmAGO2 proteins were upregulated by DA 2-3-fold at the transcriptional level. Immunoblotting is typically used for verification [23], and the dosage-dependent blot protein is the binding protein. Based on this, we confirmed that BmTudor-sn is a binding protein of DA but not BmPiwi or BmAGO2.

DARTS analysis revealed dozens of potential binding proteins, and it was necessary to determine which were the most probable. Molecular docking is a reasonable, efficient, and convenient strategy to narrow down candidates among all proteins compared to other methods such as heterologous expression and kinetic analysis, which require verification one by one. However, it seems paradoxical in this study that proteins with the top docking scores were not the binding protein [24]. Actually, docking is a completely theoretical speculation, and other experiments are required to draw a conclusion. Here, BLI kinetic analysis in vitro and CETSA in vivo were performed to address these drawbacks, demonstrating that DA binds to BmTudor-sn rather than BmPiwi or BmAGO2. BLI is a recently developed, frequently used bio-molecular interaction analysis method based on optical interference signals and is a fast, real-time method that uses a small amount of sample. Meanwhile, CETSA is a small molecule target engagement strategy based on protein thermal stability shifts caused by ligand binding, and it has been successfully used to identify several target proteins in drug studies in recent years [25]. Moreover, in the BLI analysis, we found that DA bound to BmTudor-sn at the 100 μM level, and this result agreed with the DARTS experiment in which BmTudor-sn only appeared in the 200 µg/mL treatment with a specific proteinase and not at the lower dosage and with other proteinases. Furthermore, the CETSA results demonstrated that BmTudor-sn showed thermal stability with the 200 µg/mL treatment and not at the lower dose. I2H was first used to study protein-protein interactions20, and with it, we screened the key amino acid sites of the protein-ligand complex using different mutants because of its convenience and sensitivity and because a purified protein was not required. I2H was also practical for comparing the drug inhibition rates of protein pairs.

DA has been previously reported to play a role in immune related pathways and in inducing the immunosuppression phenotype in insects [26]. The majority of studies have focused on these roles, while few have addressed the mechanism of action in cells. Therefore, previously, we assessed the ion concentration inside and outside the cell [2] and found that DA strongly induced the expression of heat shock proteins (HSPs) in cells and bound to one of them in vitro [11], and here, we showed that BmTudor-sn is a DA binding protein. Tudor-sn, a multifunctional protein, was reported to regulate downstream expression and slice RNAi under normal conditions and participate in the formation of stress granules and processing bodies under stress [15]. This may also explains why BmTudor-sn only appeared in the high concentration treatment in DARTS. Another question is whether DA binds to Tudor-sn in humans. DA is one of fungal toxins generated by the commercial biopesticide *M. anisopliae*, which also produces Destruxin B (DB), and DB was reported to have clear anti-cancer activity [27]. Interestingly, we previously showed that DB is more sensitive to V-ATPase than DA in silkworm hemocytes, and this ion channel is known to be the most probable target of DA. However, DA likely does not bind to Tudor-sn in humans because Tudor-sn has different functions in different species.

In conclusion, in vivo and in vitro experimental evidence indicated that BmTudor-sn is a binding protein of DA in silkworm Bm12 cells at the 100 µM level, and the key amino acid site of this interaction is Leu704.

## 4. Materials and Methods

### 4.1. Cell Lines and Culture

The silkworm Bm12 cell line was donated by Professor Cao Yang (College of Animal Science at South China Agricultural University) and cultured in TNM-FH culture medium (Hyclone, Pittsburgh, MA, USA) and 10% fetal bovine serum (Gibco, Waltham, MA, USA). The *Spodoptera frugiperda* 9 (Sf9) cell line was cultured in SFX culture medium (Hyclone) with 5% fetal bovine serum. Cells were cultured at 27 °C and maintained at over a period of 2–4 days. Cells in the logarithmic phase were used for the experiment.

### 4.2. Destruxin A and Treatment

Destruxin A (DA) was isolated and purified from the *Metarhizium anisopliae* var. anisopliae strain MaQ10 in our laboratory [28]. A DA stock solution of 10,000 µg/mL was made up from 1 mg of DA and 100 µL of dimethyl sulfoxide (DMSO, Sigma-Aldrich, Darmstadt, Germany). To begin treatment, the DA stock solution was added to a cell well at a final concentration of 200 µg/mL DA. The control group was only supplemented with 0.1% DMSO.

### 4.3. Homology Modeling and Molecular Docking

Homology modeling was conducted in MOE v2015.1001. The structure was determined by homology modeling of the target sequence. Template crystal structures were identified using NCBI BLAST and downloaded from the RCSB Protein Data Bank. The Protonate module of MOE v2015.1001 was used to calculate the protonation state at pH = 7. Ten independent intermediate models were built. These different homology models were obtained from the mutational selection of different loop candidates and side chain rotamers. Then, the intermediate model that scored the highest according to the GB/VI scoring function was chosen as the final model and subjected to further energy minimization using the AMBER12: EHT force field.

MOE Dock was used for molecular docking simulations. The 2D structure of the ligand was drawn in ChemBioDraw 2014 and converted to 3D in MOE v2015.1001 through energy minimization with the MMFF94x force field. The protein structures were constructed by homology modeling. The Site Finder module in MOE was used to predict the potential binding pockets. Then, the protonation state of the protein and the orientation of the hydrogens were optimized using LigX at a pH of 7 and temperature of 300 K. Prior to docking, the force field of AMBER12: EHT and the implicit solvation model of Reaction Field (R-field) were selected. The docking workflow followed the “induced fit” protocol, in which the side chains of the receptor pocket were allowed to move according to ligand conformations, with a constraint on their positions. The weight used for tethering side chain atoms to their original positions was 10. For each ligand, all docked poses were ranked by London dG scoring first, and a force field refinement was then carried out on the top 30 poses followed by a rescoring of GBVI/WSA dG.

### 4.4. Bio-Layer Interferometry (BLI)

All proteins were prepared by eukaryotic expression in the Sf9 cell line. These were tagged with His-tag and purified by nickel affinity chromatography. BLI analysis was performed on a ForteBio OctetQK System (K2, Pall Fortebio Corp, Menlo Park, CA, USA). Generally, the protein samples were coupled with a biosensor for immobilization. Serial not gradient dilutions of DA (500, 250, 125, 62.5, 31.25, 15.63, and 7.813 μM) were used for treatment. PBST buffer (0.05% Tween20, 5% DMSO) was used for the reference and dilution buffers. The working procedure was baseline for 60 s, association for 60 s, and dissociation for 60 s. Finally, the raw data were processed with Data Analysis Software (9.0, Pall Fortebio Corp, Menlo Park, CA, USA).

### 4.5. Immunoblot and Cellular Thermal Shift Assay (CETSA)

The Bm12 cell line was used to conduct the CETSA and immunoblot experiments. Cells were treated with DA. For immunoblotting, cell extracts from different DA treatments, collected after incubation with RIPA lysis buffer, were digested with proteinase K and then heated in water at 90 °C for 5 min. Samples were analyzed using SDS-PAGE and transferred to PVDF membranes. The membranes were then incubated in skim milk powder, followed by incubation with primary and HRP antibodies. ECL was added to the chemiluminescence reaction. For CETSA, Bm12 cells, treated with 200 µg/mL DA, were divided into 8 aliquots, heated at 37–58 °C, and lysed by a freeze-thaw cycle. The supernatants of the lysed cells were used for the western blot analysis as described above.

### 4.6. RNAi and Viability and Toxicity Assessment

SiRNAs were prepared by synthesis in vitro. The sequence of *BmTudor-sn* siRNA is 5ʹ-CCAAAGGACCGCCAACAAUTT-3ʹ and 5ʹ-AUUGUUGGCGGUCCUUUGGTT-3ʹ. SiRNA and FuGENE transfection reagent were each diluted in serum-free medium and then mixed. The mixture was added to Bm12 cells after the DA treatment. Viability and toxicity assessment were performed following the manufacturer’s instructions.

### 4.7. Insect Two Hybrid (I2H) System

BmTudor-sn and mutants were cloned into the I2H vector pIE-AD, and BmAGO1 was cloned into the I2H vector pIE-DBD using the Gateway system. These vectors and the luciferase vector were co-transfected into the Sf9 cell line and treated with DA at 0.02 and 0.2 µg/mL. The luciferase activities in the cell extracts were determined using a Luciferase Reporter Assay System (Promega, Beijing, China) and Synergy™ H1 (BioTek, Winooski, VT, USA).

## Figures and Tables

**Figure 1 toxins-11-00067-f001:**
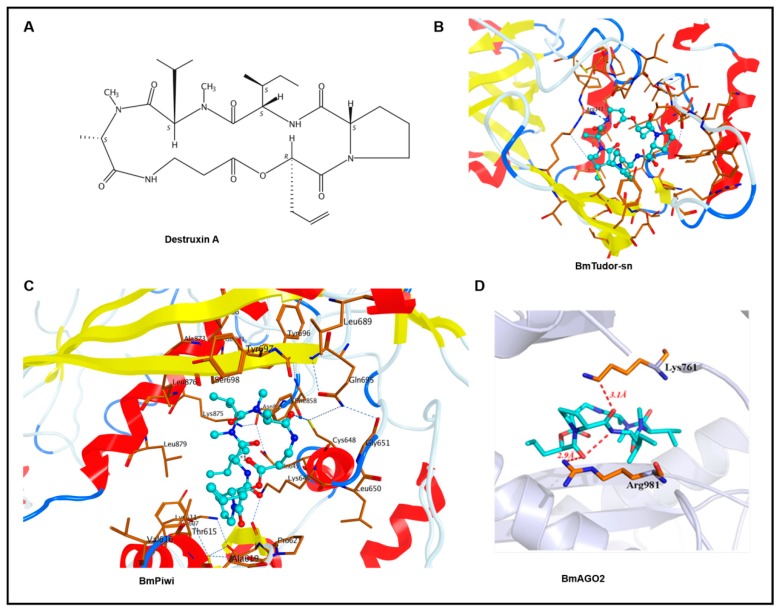
Binding pose of Destruxin A (DA) with BmTudor-sn, BmPiwi, and BmAGO2. DA is colored in cyan, and the surrounding residues in the binding pockets are colored in orange. The backbone of the receptor is depicted as spectrum ribbon. (**A**) The structure of DA. (**B**) The binding poses of DA with BmTudor-sn. (**C**) The binding mode of DA with BmPiwi. (**D**) The binding patterns of DA with BmAGO2.

**Figure 2 toxins-11-00067-f002:**
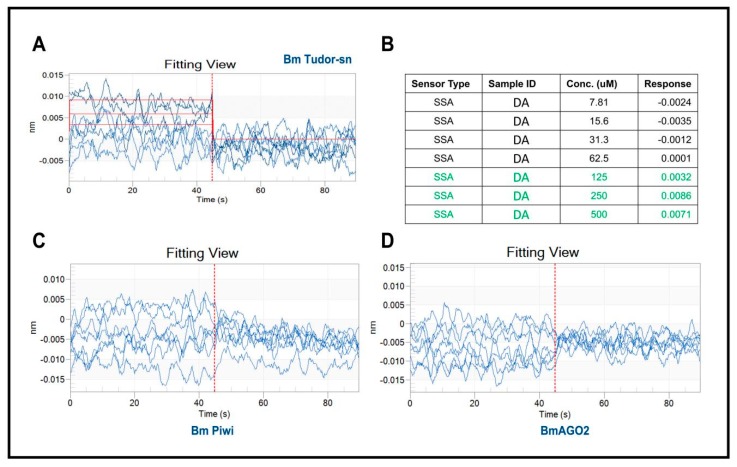
Results of recombinant proteins interacting with DA using BLI. (**A**) Data analysis with software showing there are interactions between BmTudor-sn and DA. (**B**) Molecular interaction kinetic data of BmTudor-sn with DA. (**C**,**D**) Processed data indicated no interaction of BmPiwi or BmAGO2 with DA.

**Figure 3 toxins-11-00067-f003:**
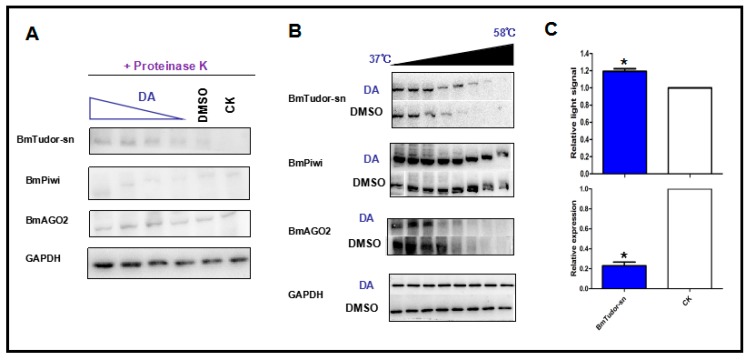
Protein stability shift assay by immunoblotting and CETSA results indicated that DA specifically binds to BmTudor-sn but not BmPiwi or BmAGO2. (**A**) Immunoblotting results demonstrating that BmTudor-sn stability was influenced by interacting with DA in a dosage-dependent manner. (**B**) CETSA showing that BmTudor-sn thermal stability was subject to thermal gradient treatment after adding DA. (**C**) BmTudor-sn knock down cells showed higher viability under DA treatment. Significant differences between columns are indicated by an * (*p* < 0.05) according to *t* test.

**Figure 4 toxins-11-00067-f004:**
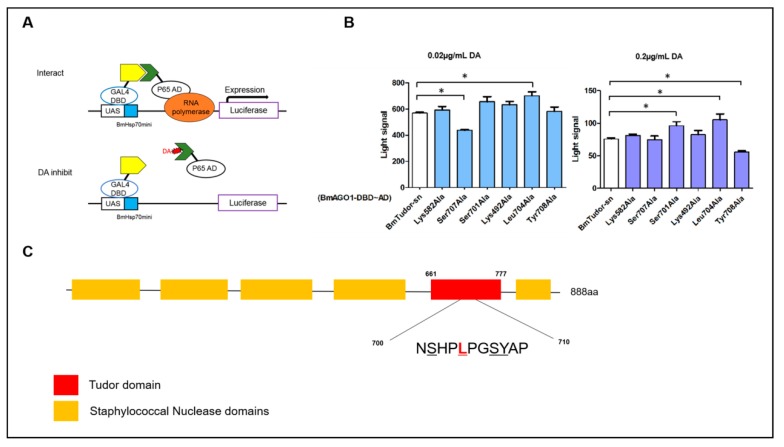
Key amino acid sites of interaction between DA and BmTudor-sn. (**A**) Schematic of the principle of the insect two-hybrid (I2H) system [19]. (**B**) Differences in the mutants with the 0.02 and 0.2 µg/mL DA treatments. Significant differences between columns are indicated by an * (*p* < 0.05) according to DMRT. (**C**) Sketch of the domain structure of BmTudor-sn and the key amino acid sites of the Tudor domain. The text continues here.

**Table 1 toxins-11-00067-t001:** The docking score of DA against BmTudor-sn, BmPiwi, and BmAGO2.

Proteins	Docking Score (kcal/mol)
BmTudor-sn	−11.002
BmPiwi	−10.8577
BmAGO2	−9.1947

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
