# Peer review of "BmTudor-sn Is a Binding Protein of Destruxin A in Silkworm Bm12 Cells"

_toxins, 2019, doi:10.3390/toxins11020067_

Round 1

Reviewer 1 Report

The experimental design and conclusions of the paper appear logical and not overstated. It appears as though the paper has been edited by an English speaker who is not familiar with scientific terminology. For this reason the paper appears to be grammatically ok, but there are sections with incorrect scientific terminology and the wrong descriptive words used, making it difficult to understand the content. Take for instance line 96-97 the sentence implies that recombinant proteins and lepidoptera Spordoptera frugiperda cells have similar folding. In the same sentence, the proteins were not “obtained”, they were most likely expressed and purified, and the cell line used because proteins expressed in an insect cell line are believed to have a more similar tertiary structure (including protein folding and have similar protein modifications such as glycosylation etc) to the natively expressed protein. I would recommend that the paper is edited by a scientific editor.

Figures should describe how the experiments were completed rather than describing the results.

Molecular docking results – more complete results could be shown in supplementary data.

Furthermore, it is completely unclear how  and why the three proteins were selected. In the introduction the first paragraph talks of the immunosuppression of DA in insects, and their previous results identifying heat shock proteins. Then for some reason the focus switches to germ cell development and the Argonaute family of proteins.

Figure 1. Recommend to change panels B and D to be the similar to paned C.

Line 77. Hydrogen bonds are not a type of hydrophobic interactions.

Section 2.2  - Needs to be completely re-written, and the last two sentences removed. More detail about the results required.

Figure 2. Panel A and D are the same. Panel B should summarise all results, only the last two columns are informative. The timings on the plots do not agree with the methods so it is difficult to understand what is the binding and not the binding. No base line shown.

Line 159. 7 should be superscript, likewise for line 191

The discussion is mainly very well written, the first paragraph needs a lot of work and could either be moved to the end of the discussion section or removed altogether. The second paragraph is well written and an appropriate place to start the discussion.  

Line 188 DA has been previously reported….

Line 195 This may also explain…

Line 197 M. anisophliae should be in italics.

Line 203 in vivo and in vitro

More details are required in the methods sections.

In particular

How was destruxin A purified. If this has been previously published in full state –following previously published methods …, were there modifications to the previous method. It is also good to state briefly what method was used.

How were the homology models constructed.

Full details of genes used and cloning and expression.

Line 244. Serial not gradient dilution

References appear to have the journal short hand after the author list. All references should be corrected in the appropriate style.

Author Response

Dear reviewer,

Thanks for advisable suggestion for manuscript. Point-by-point response in PDF file. 

Reviewer 2 Report

In the present study, the Authors examined the interaction of destruxin A with BmPiwi, BmAGO2, and BmTudor-sn proteins. The topic of the investigation is interesting, and the manuscript was written in a good linguistic and scientific style. The experimentation seems appropriate, and the discussion is consistent with the results. Based on my opinion, some modifications are reasonable, therefore, I suggest the publication of the manuscript after a minor revision. My comments are listed below.

Minor linguistic correction of the text is necessary (e.g., lines 25-26).

The binding or association constant of the formed protein-ligand complex need to be quantified. It would be very interesting to know.

A more detailed description/explanation of methods applied need to be inserted into the text.

Cytotoxicity and viability assays as well as their demonstration in Fig. 3C in unclear for me. Please provide a more detailed description and explanation.

It would be better to represent Fig. 3C as a separate figure.

Resolution of figures should be improved. Font size in Figs. 2-4 is too small.

What kind of statistical analysis was performed? Details should be inserted into the Mat&met section. Fig. 3C does not contain statistics.

Author Response

Dear reviewer,

Thanks for advisable suggestion for manuscript. Point-by-point response in Word file.
